# Expression of EMT-Related Genes in Hybrid E/M Colorectal Cancer Cells Determines Fibroblast Activation and Collagen Remodeling

**DOI:** 10.3390/ijms21218119

**Published:** 2020-10-30

**Authors:** Irina Druzhkova, Marina Shirmanova, Nadezhda Ignatova, Varvara Dudenkova, Maria Lukina, Elena Zagaynova, Dina Safina, Sergey Kostrov, Dmitry Didych, Alexey Kuzmich, George Sharonov, Olga Rakitina, Irina Alekseenko, Eugene Sverdlov

**Affiliations:** 1Research Institute of Experimental Oncology and Biomedical Technologies, Privolzhsky Research Medical University, 603005 Nizhny Novgorod, Russia; danirin@yandex.ru (I.D.); shirmanovam@gmail.com (M.S.); n.i.evteeva@gmail.com (N.I.); orannge@mail.ru (V.D.); kuznetsova.m.m@yandex.ru (M.L.); ezagaynova@gmail.com (E.Z.); sharonov@gmail.com (G.S.); 2Lobachevsky State University of Nizhny Novgorod, 603950 Nizhny Novgorod, Russia; 3Department of Molecular-Genetic Basis of Biotechnology and Protein Engineering, Institute of Molecular Genetics of National Research Centre «Kurchatov Institute», 123182 Moscow, Russia; nauruz@mail.ru (D.S.); kostrov@img.ras.ru (S.K.); irina.alekseenko@mail.ru (I.A.); edsverd@gmail.com (E.S.); 4Department of Genomics and Postgenomic Technologies, Shemyakin-Ovchinnikov Institute of Bioorganic Chemistry of The Russian Academy of Sciences, 117997 Moscow, Russia; dmitrydid@gmail.com (D.D.); rakitinaolga97@gmail.com (O.R.); 5Institute of Translational Medicine, Pirogov Russian National Research Medical University, 117997 Moscow, Russia; 6Laboratory of Epigenetics, FSBI «National Medical Research Center for Obstetrics, Gynecology and Perinatology named after Academician V.I. Kulakov» Ministry of Healthcare of the Russian Federation, 117198 Moscow, Russia; 7National Research Center «Kurchatov Institute», 123182 Moscow, Russia

**Keywords:** cancer-associated fibroblasts, colorectal cancer cells, collagen remodeling, fibroblast activation, epithelial/mesenchymal state

## Abstract

Collagen, the main non-cellular component of the extracellular matrix (ECM), is profoundly reorganized during tumorigenesis and has a strong impact on tumor behavior. The main source of collagen in tumors is cancer-associated fibroblasts. Cancer cells can also participate in the synthesis of ECM; however, the contribution of both types of cells to collagen rearrangements during the tumor progression is far from being clear. Here, we investigated the processes of collagen biosynthesis and remodeling in parallel with the transcriptome changes during cancer cells and fibroblasts interactions. Combining immunofluorescence, RNA sequencing, and second harmonic generation microscopy, we have explored the relationships between the ratio of epithelial (E) and mesenchymal (M) components of hybrid E/M cancer cells, their ability to activate fibroblasts, and the contributions of both cell types to collagen remodeling. To this end, we studied (i) co-cultures of colorectal cancer cells and normal fibroblasts in a collagen matrix, (ii) patient-derived cancer-associated fibroblasts, and (iii) mouse xenograft models. We found that the activation of normal fibroblasts that form dense collagen networks consisting of large, highly oriented fibers depends on the difference in E/M ratio in the cancer cells. The more-epithelial cells activate the fibroblasts more strongly, which correlates with a dense and highly ordered collagen structure in tumors in vivo. The more-mesenchymal cells activate the fibroblasts to a lesser degree; on the other hand, this cell line has a higher innate collagen remodeling capacity. Normal fibroblasts activated by cancer cells contribute to the organization of the extracellular matrix in a way that is favorable for migratory potency. At the same time, in co-culture with epithelial cancer cells, the contribution of fibroblasts to the reorganization of ECM is more pronounced. Therefore, one can expect that targeting the ability of epithelial cancer cells to activate normal fibroblasts may provide a new anticancer therapeutic strategy.

## 1. Introduction

It is known that the development and progression of malignant tumors, as well as their therapeutic resistance, depends not only on the intrinsic characteristics of the cancer cells but also to a significant degree on the tumor stroma [1,2]. A better understanding of the tumor–stroma interaction mechanisms is important in respect of the search for novel prognostic markers and therapeutic targets. The majority of experimental data demonstrate the improvements of therapeutic outcomes due to the application of stroma-targeting approaches, especially combined with traditional therapies [3]. However, there are no completely curative strategies and some of them, particularly in case of lysyl oxidase homolog 2 LOXL2 inhibitors, have failed clinical trials [3,4]. Taken together, it says that our knowledge in this field is insufficient, and the search for novel stromal targets should be carried out.

Among the stromal components, cancer-associated fibroblasts (CAFs) supposedly play an important role in the formation of the specific tumor microenvironment. CAFs represent a population of activated fibroblasts within tumors that is similar, in many ways, to the myofibroblasts involved in wound healing. It is believed that most CAFs derive from the activation of local tissue-resident fibroblasts, although the conversion of several other cell types (e.g., adipocytes, pericytes, endothelial cells) into CAFs is also documented [5]. The key differences of CAFs from normal (quiescent, non-activated) fibroblasts are their enhanced proliferation and biosynthetic activity, such as the production of extracellular matrix (ECM) components, remodeling enzymes, cytokines, and growth factors. Determining the molecular markers for the identification of CAFs is still an ongoing area of investigation [6]. The expression of the most commonly used markers—α-smooth muscle actin (aSMA), fibroblast-activating protein (FAP), and platelet-derived growth factor receptors A/B (PDGFRa/b)—varies strongly among CAF subpopulations, reflecting the highly heterogeneous nature and plasticity of CAFs.

The exact mechanisms by which fibroblasts become activated in a tumor remain largely unknown. In the case of wound repair, the activation of fibroblasts is initiated by chemical mediators released by the damaged epithelial cells and by the immune cells recruited to the damage site. With respect to CAFs, the activation can be caused by additional chemical mediators produced by the cancer cells.

In a tumor, there is permanent crosstalk between the cancer cells and fibroblasts in which the cancer cells induce and support the activated phenotype of the fibroblasts, while the fibroblasts produce and remodel the ECM and stimulate the migratory activity of the cancer cells. As one of the results of such cooperation, the cancer cells acquire mesenchymal features, which can then contribute to their active migration and metastasis. This acquisition is associated with changes in the expression of genes involved in both intracellular and extracellular processes. In particular, the composition and presentation of surface adhesion molecules might change, as well as the activity of extracellular matrix reorganization and structuring processes.

Collagens are the major proteins of the ECM. In solid tumors, collagen is often cross-linked and linearized, which increases the tissue stiffness and provides paths for directed cancer cell invasion [7]. Previously, the desmoplastic stromal reaction characterized by the deposition of fibrillar collagens (primarily types I and III) and fibroblasts around the tumor was considered to be a response of the host tissue to the presence of invasive tumor cells, forming a barrier against the expansion of the cancer cells [8]. Recent studies suggest that desmoplasia, on the contrary, forms favorable conditions for tumor progression and metastasis via modulation of the biological behavior of the tumor cells, including their gene expression, proliferative activity, adhesion, migration, and apoptosis [9,10,11].

It has been shown that alterations in the microenvironment induced by CAFs (e.g., matrix stiffness) can initiate the phenotypic transformation of cancer cells from an epithelial to a mesenchymal state (the epithelial–mesenchymal transition, EMT). Nevertheless, the intrinsic epithelial/mesenchymal states of the cancer cells still remain an important determinant of the tumor–stroma co-evolution. The exact role of each of these components is still to be determined.

Another unanswered question is the role of cancer cells in organizing the collagen. Collagen synthesis and remodeling is classically regarded as one of the basic functions of activated fibroblasts in both normal and cancerous tissues. However, there are data showing that cancer cells, themselves, are capable of promoting contraction, cross-linking, and degrading collagen [12].

By now, the participation of the stroma in the tumor progression is most extensively investigated for breast cancer [13,14,15,16]. As for colorectal cancer, which is the third most common cancer type, there are only scattered data about the specifics of stroma organization [7,9,10,11]. The majority of the studies are focused on the regulation of the disease though the CAFs signaling, while the ECM is much less explored [17,18]. In this study, we sought to elucidate the correlations between the epithelial/mesenchymal states of colorectal cancer cells and their ability to activate normal fibroblasts and ECM remodeling. To this end, we exploited human colorectal cancer cell lines with different epithelial/mesenchymal states co-cultured with normal fibroblasts in a 3D collagen matrix. Patient-derived CAFs and colon tumor xenografts in mice were used to validate the in vitro findings. Using immunofluorescence assay, RNA-sequencing, and second harmonic generation (SHG) microscopy, we demonstrated phenotypic, transcriptomic, and functional changes in the fibroblasts upon their activation by cancer cells, and this correlated with the intrinsic migratory capacity of the cancer cells.

## 2. Results

### 2.1. Epithelial/Mesenchymal States of Cancer Cells

In a search for the cellular mechanisms underlying the interaction of cancer cells and normal fibroblasts, we examined various colorectal cancer cell lines for their epithelial/mesenchymal states. The well-described model cell lines HT29, HCT116, SW480, Caco-2, and SW837 were chosen for our experiments based on their ATCC description as cell lines with different epithelial/mesenchymal morphologies.

The wound-healing assay showed that the HT29 and SW837 cell lines had the least migratory activity (Figure 1A). Caco-2 healed wounds more quickly, but the process was not completed by 72 h. HCT116 and SW480 showed an enhanced rate of wound closure compared with the other three cell lines.

It is known that the loss of ability to form spheroids correlates with increased migration and invasive activity of the cells [19]; therefore, we analyzed the cell lines for spheroid formation capacity (Figure 1B). It was found that two cell lines (SW480 and Caco-2) did not form cell aggregates by 7 days after seeding into ultra-low attachment round-bottom plates. The other three cell lines formed compact multicellular spheroids, but in the cases of HCT116 and SW837, a small proportion of the cells did not integrate into the spheres and was visible at the border.

Vimentin expression, loss of E-cadherin, enhanced migratory activity, ECM production, and invasiveness accompany the acquisition of mesenchymal features by cancer cells. We assessed the expression of E-cadherin and vimentin, which are two key markers of the epithelial-to-mesenchymal transition (EMT), using immunofluorescence staining. The E-cadherin level was the highest in HT29 cells and lowest in HCT116. The SW480, Caco-2, and SW837 cell lines displayed a moderate expression. Vimentin expression was detected only in the SW480 cells.

These data characterize HT29 as epithelial cells with a low level of migration activity and SW480 as the line with the highest migratory capacity and a mesenchymal phenotype.

To investigate the molecular mechanisms underlying the observed difference in the epithelial/mesenchymal states of the HT29 and SW480 cell lines, RNA-seq and subsequent transcriptomic comparisons were performed. The analysis (Appendix A) revealed 2044 differentially expressed genes (DEGs) in HT29 cells (that is, genes with *p-adj* <0.05; TPM (in HT29) ≥5 and log2(fold change, FC) ≤−1) and 2152 SW480 DEGs (that is, genes with *p-adj* <0.05; TPM (in SW480) ≥5 and log2(FC) ≥1). Among these genes, Gene Ontology-based functional enrichment analysis revealed 248 and 389 DEGs involved in cell migration and adhesion in the HT29 and SW480 lines, respectively (Appendix A). Cell migration and adhesion are known to be closely related processes [20]. We compared the Gene Ontology Biological Process (GO BP) terms related to cell migration and adhesion, for which significant (*p-adj* <0.05) enrichment was found in the studied HT29 and SW480 DEGs. In general, the enrichment in categories related to migration and adhesion is more typical for the SW480 DEGs (Figure 1C), thus confirming the observed phenotypic difference of these cell lines (Figure 1A,B). HT29 DEGs are enriched in only a few categories, those related to epithelial cell migration and collective migration (“tissue migration”), as shown in Figure 1C. These observations correspond to the observed increased mobility and invasiveness of SW480 cells compared to HT29 cells (Figure 1A,B).

Despite the greater activity of their cell-adhesion-related genes, SW480 cells were not able to form spheroids, in contrast to HT29 cells. This phenomenon might be explained by the low expression level in SW480 cells of the E-cadherin gene (CDH1), which is one of the key cell-contact and spheroid-formation-related proteins, [19,21].

We have also studied epithelial and mesenchymal marker genes [22,23,24,25], which were differentially expressed in HT29 and SW480 (Appendix A, respectively). Among ten differentially expressed epithelial markers, four genes have higher expression in SW480 and six genes, including *CDH1* and *KLF5*, have higher expression in HT29 cells (Figure 1E (left)). In addition, 16 differentially expressed mesenchymal marker genes were identified, 11 of which have higher expression in SW480 (including *ITGA5*, *VIM*, *CTNNB1*, *MMP14*, *FN1*, *MMP9*, and *SNAI1*), and five have higher expression in HT29 cells (Figure 1E (right)). The expression of *EPCAM* was similar in HT29 (237 TPM) and SW480 (255 TPM) cells, which corresponds well with the data from immunofluorescent staining (Figure 1D). The observed combination of epithelial and mesenchymal features in the SW480 cells demonstrates their hybrid phenotype, which is significantly biased to the mesenchymal, when compared to HT29 cells, thus confirming the difference between these cell lines as observed with immunofluorescent staining (Figure 1D).

Thus, the ICH staining, the transcriptome investigation, and the functional tests all confirmed the epithelial phenotype with low migratory capacity of the HT29 cell line and the mesenchymal phenotype with high migratory activity of the SW480 cell line.

### 2.2. The Molecular Phenotype of NFs and CAFs and Activation of Normal Fibroblasts in Co-Culture

We performed immunofluorescence analysis for two major CAF markers: FAP and aSMA in normal fibroblasts (NFs) and CAFs, which were isolated from the patients’ colorectal tumors. Despite the FAP being considered as a CAF marker, in our experiments, the FAP expression in CAFs and NFs was nearly the same, while aSMA, which is typical of myofibroblasts, had a significantly higher expression level in the CAFs (Figure 2).

RNA-seqs of the NFs and two cultures of CAFs (CAF1 and CAF2) were performed to confirm the observed difference. Comparison of the expressions of 24 CAF marker genes (Appendix A; [26,27,28,29,30,31,32,33]) in the NFs with those in CAF1 and CAF2 revealed six genes, whose expression in CAF1 and CAF2 was considerably (>2 times) higher than in the NFs (*ACTA2, ASPN, CXCL10, PDPN, THY1*, and *TNC*). In addition, the *POSTN* expression level was more than 2-fold higher in CAF1 than that in the NFs (*p-adj* < 0.05; Figure 2E), and also, the TPM for this gene was >5 in CAF1. Thus, despite FAP expression being similar in the NFs and CAFs (Figure 2E), the activity of other known CAF markers was significantly higher in both isolated CAF lines compared to that in the NFs.

We simulated in vitro the NFs’ interactions with cancer cells by the co-cultivation of NFs with the cancer cell lines, HT29, HCT116, and SW480, of which HT29 was preliminarily identified as the least invasive, and SW480 was preliminarily identified as the most invasive cell line.

With immunofluorescence assay, an elevated FAP expression was observed in the NFs of all three co-culture systems, starting at day 2 of culturing, with the maximum being in co-culture NF+HCT116. After that, in the case of NF+HCT116 and NF+SW480 the co-cultures, FAP expression dropped. In the case of NF+HT29 co-culture, the maximum FAP expression was achieved at day 5, which was followed by a slight decrease by day 7. A similar dynamic was observed for aSMA expression with a sharp increase by day 2, which was followed by gradual decrease by day 5 in all three co-cultures and a dramatic drop to the level registered in NFs in the NF+HCT116 and NF+SW480 co-cultures by day 7. At day 1 in NF+SW480 co-culture, the aSMA expression was even lower than in the NFs, but it was accompanied by the maximal FAP expression for this co-culture. These data demonstrate the activation of normal fibroblasts in the presence of cancer cells, which is clearly seen from the expression of aSMA. An increase in FAP expression in the co-cultures was also observed, although normal skin fibroblasts originally had quite a high level of FAP, which is consistent with their physiological functions.

It is notable that cancer cells in the co-cultures expressed high levels of FAP and aSMA, but their further analysis was out of the scope of this study.

Transcriptome analysis of NFs before and after 5 days of co-cultivation with the HT29 (NF^HT29^) and SW480 (NF^SW480^) cell lines was performed to confirm the observed fibroblast activation. We compared the expression of the seven CAF marker genes mentioned above (*ACTA2, ASPN, CXCL10, PDPN, THY1, TNC,* and *POSTN*) in these cells and revealed a statistically significant (about 2-fold, *p-adj* < 0.05) increase in *ACTA2, POSTN,* and *TNC* gene expression as a result of co-cultivation with both HT29 and SW480 cells (Figure 2E). The expression level of all three genes in both NF^HT29^ and NF^SW480^ was significantly lower than that in CAFs. We also found that the *DES* gene (one of the 24 CAF marker genes) expression level was more than 2-fold higher in both NF^HT29^ and NF^SW480^ compared to NFs alone, its expression level being similar in CAFs and NFs in pure culture. The activation of the expression of *ACTA2* (2.1-fold in NF^HT29^ vs. 1.3-fold in NF^SW480^) and *DES* (4.9-fold in NF^HT29^ vs. 3.4-fold in NF^SW480^) was more significant in NF^HT29^, the expression of *POSTN* was similarly activated in NF^HT29^ (1.6-fold) and NF^SW480^ (1.7-fold), and the activation of the expression of *TNC* (1.6-fold in NF^HT29^ vs. 2.2-fold in NF^SW480^) was more significant in NF^SW480^. Thus, the RNA-seq data might indicate that during co-cultivation, HT29 cells activate normal fibroblasts to a greater extent than do SW480s, which corresponds well with the immunofluorescence staining data. The *ACTA2* expression enhancement as a result of co-cultivation correlates well with the results of aSMA immunofluorescent staining (Figure 2C,D). In addition, RNA-seq confirmed the similarity of *FAP* expression in the NF and NF^SW480^ cells (Figure 2C–E). However, the RNA-seq data were not consistent with the observed immunofluorescence enhancement of FAP staining in NF^HT29^ after 5 days of co-cultivation (Figure 2C–E). This result might be explained by the observed decreases in FAP expression during co-cultivation (Figure 2C,D). The decreases might be caused by the suppression of *FAP* expression by the 5th day of co-cultivation, leading to the similar detected levels of *FAP* transcript in NFs and NF^HT29^. Meanwhile, the amount of surface FAP in NF^HT29^ detected at this point by immunofluorescence staining had not yet dropped.

Therefore, co-cultivation of NFs with cancer cells resulted in a moderate activation of individual CAF gene markers that allows us to conclude that activated fibroblasts at the moment of recruitment by cancer cells represent a bridging form of myofibroblasts rather than mature CAFs.

### 2.3. Collagen Remodeling

#### 2.3.1. Collagen structure and Collagen-Related Gene Expression in Monocultures of Cancer Cells

The ability of three colorectal cancer cell lines (HT29, HCT116 and SW480) to remodel collagen was studied in collagen-based 3D models using SHG microscopy. It was found that cancer cells, when cultured alone, differ in their capacity to remodel collagen (Figure 3A); all quantitative parameters of the SHG signal and *p*-values are given in Appendix A. Overall, in monocultures of cancer cells, fibrillar collagen was present in low quantity. An increase in the density of collagen was observed on day 5 of cultivation only for two invasive cell lines, HCT116 and SW480. Of the three cell lines, SW480 demonstrated the ability to create quite thick, oriented fibers starting from day 5, which correlates with its more mesenchymal phenotype. This effect is consistent with data on the expression of the genes involved in collagen production and remodeling (see below).

Collagen-related gene expression was assessed in HT29 (epithelial phenotype) and SW480 (mesenchymal phenotype) cells using RNA-seq (Figure 4A,B). The expression of genes, related to collagen biosynthesis and positive regulation (B+PR); collagen fibril organization and ECM aggregation (CFO+EA); and collagen catabolism and negative regulation (C+NR) was estimated. The genes were included in the gene categories according to GO BP terms with manual correction and [34], and the gene lists are shown in the Appendix A, respectively. Venn diagrams and histograms were used to compare the genes significantly expressed in HT29 and SW480 (*p-adj* < 0.05; TPM ≥ 5) and show that more genes, related to collagen biosynthesis and positive regulation, and to collagen catabolism and negative regulation, are active in cell line SW480 compared to HT29 (Figure 4A,B). In spite of the fact that more genes, related to collagen fibril organization and ECM aggregation, are active in cell line HT29 compared to SW480, two important collagen-remodeling genes, LOXL2 and LOXL3 (lysyl oxidases), are more active in SW480 compared to HT29 (Figure 4B). The observed differences may indicate that SW480 cells interact with collagen much more actively than do HT29, and this corresponds well with the SHG microscopy data (Figure 3A).

#### 2.3.2. Collagen Remodeling and Collagen-Related Gene Expression in Co-Cultures of Cancer Cells and Normal Fibroblasts

In the presence of fibroblasts, the collagen structure significantly changed (Figure 3B); all quantitative parameters of the SHG signal and *p*-values for co-cultures are given in Appendix A. Co-culturing of all cancer cell lines with normal fibroblasts resulted in a statistically greater amount of fibrillar collagen during all periods of cultivation. Notably, the deposition of collagen was more pronounced in the case of the more invasive cells, HCT116 and SW480, where a 2-fold increase in the collagen density, compared with the monocultures, could be observed after 1–2 days of co-culturing. In the case of HT29, such an increase in the collagen density was achieved after 7 days. In the process of co-culturing, the fibers become more heterogeneous in size. Furthermore, a co-culture with HT29 formed an increased number of large fibers, as compared with the monocultures. The high proportion of thick fibers in the case of co-culture with SW480 may be attributed to the invasiveness of this cell line itself, rather than to its interaction with the fibroblasts. The most noticeable differences in the matrix architecture resulting from the interaction of cancer cells and fibroblasts were observed for the ordering of the collagen fibers. In all co-cultures, the fibers become more organized and more aligned with time, and particular orientations of the fibers could be distinguished. Note that such ordering of the fibers is a relatively early process during the interaction of the cancer cells and NFs; already after 1 day of co-culturing, the parameter of coherency exceeded that in the cancer cell monocultures and in pure collagen gel (≈0.02 a.u).

Comparison of the collagen structures formed by NFs in monoculture (Figure 3C) and in the co-cultures revealed a greater collagen content and a more uniform structure of the collagen in the NFs monocultures. That is, the ECM in the NFs culture was represented by a large number of collagen fibers with equal size and uniform distribution, resulting in a dense homogeneous structure of the ECM, while in co-cultures, the ECM was less dense with non-uniformly distributed collagen fibers.

RNA-seq of HT29 and SW480 cancer cells and NFs before and after 5 days of co-cultivation was performed to compare the effects observed with the changes in gene expression patterns. The expression of collagen-related genes was estimated. Figure 4C shows the distributions of the log2(FC) value for significantly expressed collagen-related genes in three gene categories (B+PR, CFO+EA, and C+NR); that is, the expression of the gene in co-culture divided by the expression of the gene in monoculture. The shift of the median of the log2(FC) for genes related to collagen fibril organization and ECM aggregation toward co-cultured cancer cells (both HT29 and SW480) may indicate the activation of collagen remodeling processes in these cells as a result of the co-cultivation. Likewise, the shift of the median of the log2(FC) for genes related to collagen catabolism and negative regulation toward cancer cells before co-cultivation (both HT29 and SW480) may indicate the suppression of collagen catabolic processes in these cells as a result of co-cultivation. The observed shifts of the median of log2(FC) for collagen biosynthesis and positive regulation genes category are too small to cause significant phenotypic effects.

In the case of fibroblasts, co-cultivation with both cancer cell lines tended to increase the expression of the genes related to collagen biosynthesis, collagen fibril organization, and ECM aggregation. In turn, the expression of genes related to collagen catabolism and negative regulation in the fibroblasts is slightly down-regulated. In general, the number of significantly expressed collagen-related genes was greater in the NFs compared to that in the cancer cells in both co-cultures (Figure 5A), which may indicate that NFs are the main collagen-remodeling component of the co-cultures.

To reveal differences between the co-cultures, we have also compared the significantly expressed collagen-related genes in co-cultivated cancer cells (HT29^NF^ vs. SW480^NF^) and fibroblasts (NF^HT29^ vs. NF^SW480^), as shown in Figure 5. The main difference is observed between the cancer cells (Figure 5, Appendix A), with the majority of differing genes being most evident between the HT29 and SW480 monocultures (Figure 4B). Despite the small number of collagen-related genes, which determine the difference between NF^HT29^ and NF^SW480^, it can be noted (Figure 5B, Appendix A) that two genes from the collagen biosynthesis and positive regulation group, *COL7A1* and *COL5A3*, were highly active in NFs monoculture, and they increased their activity only during co-cultivation with SW480. The *MMP3* gene responsible for collagen proteolysis was also expressed at a high level in the NFs monoculture and down-regulated only during co-cultivation with SW480.

Therefore, the observed up-regulation of the expression of the genes related to collagen fibril organization and ECM aggregation in both cancer cells and NFs and also the up-regulation of the expression of the genes related to collagen biosynthesis and positive regulation in NFs might explain the observed by SHG microscopy formation of the dense ECM, consisting of large, highly oriented collagen fibers (Figure 3B) as a result of co-cultivation. The observed difference in collagen remodeling between two co-cultures is most likely explained by the different collagen remodeling activities of the cancer cells present and not by any different collagen remodeling activity of the fibroblasts.

#### 2.3.3. Collagen Organization and Collagen-Related Gene Expression in CAFs and Normal Fibroblasts

A comparison of patient-derived CAFs and normal fibroblasts revealed strong differences in their ability to remodel collagen (Figure 3C); all quantitative parameters of the SHG signal and *p*-values for CAFs and NFs are given in Appendix A. Both types of cell culture formed dense collagen networks within 2 days; subsequently, however, in the culture of CAFs, the collagen density gradually decreased, while in the culture of normal fibroblasts, it did not change. In the culture of CAFs the number of collagen fibers was gradually reduced during cultivation, and only the larger collagen bundles were preserved, while in the culture of normal fibroblasts, a marked increase in the number and enlargement of collagen fibers were observed by day 7. In both cases, the collagen was quite highly oriented (i.e., highly ordered), and the degree of ordering varied insignificantly during 7-day cultivation. Consequently, the major difference between CAFs and normal fibroblasts is a predominance of proteolytic processes in the CAFs culture, against which individual large collagen fibers are preserved, while normal fibroblasts tend to form a dense fibrous structure containing uniform fibers.

We have also compared the expression of three groups of collagen-related genes in NFs and CAFs (CAF1 and CAF2) (Appendix A). The log2(FC) median of the B+PR gene group is slightly shifted toward normal fibroblasts compared to CAF1, which may indicate that collagen biosynthetic processes are more active in NFs than in CAF1.The other medians are not significantly shifted. These results support the SHG microscopy observations (Figure 3C), that is, a gradual increase in the number of collagen fibers during the cultivation of NFs, and proteolysis of collagen fibers during the cultivation of CAFs. Thus, compared to CAF1, normal fibroblasts might more actively participate in the processes of collagen biosynthesis.

Therefore, using SHG-based microscopy, we identified specific changes in the collagen morphology that develop when fibroblasts interact with cancer cells in vitro. Fibroblast-mediated remodeling of collagen leads to the formation of a highly ordered fiber network, which is more favorable for cancer cell invasion. Furthermore, the most invasive cancer cells, themselves, are capable of remodeling collagen to form oriented bundles and demonstrate the corresponding gene activity. Similar structural features of collagen were also typical of cultured CAFs isolated from patients’ tumors. However, under prolonged cultivation in the absence of cancer cells, the CAFs lost the ability to organize collagen in ways to promote invasion and initiated matrix degradation. RNA-seq analysis demonstrated the elevated activity of all collagen-related genes in fibroblasts compared to cancer cells, confirming the predominant role of fibroblasts in collagen remodeling. Furthermore, the RNA-seq analysis, by revealing an increased activity of the collagen biosynthesis and organization genes in NFs compared to CAFs, suggests that NFs recruited by cancer cells are necessary participants in the tumor invasion front.

#### 2.3.4. Collagen Structure in Tumor Xenografts

To determine if the innate epithelial/mesenchymal states of cancer cells influences collagen fiber organization in vivo, we inoculated HT29, HCT116, and SW480 cells subcutaneously into mice to generate tumors and assessed the resulting collagen properties using SHG microscopy.

Of these three cell lines, HT29 and HCT116 displayed high susceptibility (four of four tumors of each type) and similar growth rate in mice, reaching ≈700 mm^3^ in two weeks after inoculation. HT29 tumors had slightly slower growth rate in mice than HCT116 (≈180 mm3 vs. ≈400 mm3) in two weeks after inoculation (Figure 6B). A SW480 tumor developed only in one of four animals and was much smaller in size, ≈50 mm^3^, and therefore, it was not considered. In spite of the different growth rate, histologically, both HT29 and HCT116 tumors had a dense, cellular structure with a moderate amount of stroma; the tumors contained a small number of necrotic cells, and the blood vessels were present mainly in the periphery (Figure 6A).

Therefore, a comparison of collagen structure in vivo was made for the HT29 and HCT116 tumors. We found that compared to HCT116, HT29 formed a denser stroma with more non-uniform and more oriented collagen fibers (Figure 6C,D). This observation was confirmed by the quantitative metrics. The parameters of the SHG signal, such as density, energy, and coherency showed statistically significant differences (*p* < 0.00001) between these tumor types.

The collagen organization in vivo in the HCT116 and HT29 xenografts results are consistent with the capacity of these two cell lines to activate normal fibroblasts, as was shown by immunofluorescence staining for FAP and aSMA in vitro. The activation of fibroblasts in the presence of HT29 cells was more excessive and prolonged than in the case of HCT116. This result allows us to suggest that cancer cells with a more-epithelial phenotype stimulate fibroblasts to a greater extent, ensuring a fibrous microenvironment.

## 3. Discussion

This work demonstrates the links between the epithelial/mesenchymal states of colorectal cancer cells, their ability to induce the transformation of normal fibroblasts into CAFs, together with the dynamics of this process and the organization of collagen. For the first time, communication between cancer cells and normal fibroblasts has been investigated at the molecular, phenotypic, and functional levels within one study. RNA profiling was performed to provide insight into the molecular mechanisms related to fibroblast activation and remodeling of the extracellular matrix.

### 3.1. Communication of Colorectal Cancer Cells and Normal Fibroblasts Induces Activation of the Normal Fibroblasts

Previously, numerous studies have demonstrated close interactions between different tumor components: cancer cells, the extracellular matrix, and non-cancerous cells associated with tumors, among which CAFs are the most abundant type of cells [35]. Usually, CAFs are described as activated cells with myofibroblast-like morphology distinguishable from that of resident tissue fibroblasts by their elevated production of cytokines, chemokines, metabolites, enzymes, and extracellular matrix components [36]. The precise mechanism of their emergence has not yet been determined [37], although resident quiescent fibroblasts are considered to be the main source from which CAFs originate [36,38,39]. Despite the abundant list of potential markers of fibroblast activation [36,40], there are no unique characteristic for all CAFs. Markers can differ depending on tumor type and may dynamically change during cancer progression, likely reflecting the CAFs’ plasticity [37].

Obtaining colon fibroblasts from healthy donors is arduous; therefore, to investigate the activation process of normal fibroblasts by cancer cells, we used skin fibroblasts from healthy donors. These reportedly, upon activation, have similar phenotypic features to myofibroblasts from skin wounds, those from pathological fibrotic tissue, and the fibroblasts in and around epithelial tumors [41]. It has been demonstrated that breast cancer cells activate normal dermal and mammalian fibroblasts in the same way [42], with the effects of dermal and colon fibroblasts on colon cancer cells being quite similar [43]. It has also been shown that HT29 cells activate normal colon fibroblasts in a way similar to our results in terms of aSMA expression and the elevation of ECM proteins [44]. In addition, the formation of highly ordered collagen structures is typical of all known pathological processes where myofibroblasts act as major participants [45,46]. So, we assume that skin fibroblasts can adequately model the processes typical of resident colon fibroblasts interacting with cancer cells in respect of ECM remodeling and cancer specific activation.

We used two major CAF markers—FAP and aSMA—to assess the dynamic changes in the NFs during co-cultivation with three colorectal cell lines. The activation of normal fibroblasts was clearly demonstrated by increases in the expression of FAP and aSMA at the beginning of their interaction with the cancer cells, although the expression of both markers decreased after further co-cultivation. Notably, the FAP expression demonstrated a sharp decline in co-cultures with the more-invasive cancer cells (HCT116 and SW480) but a gradual decrease in the presence of less-invasive cancer cells (HT29). In general, the aSMA expression had the same dynamics.

Data on the direct activation of fibroblasts by cancer cells are scarce, which is likely due to the absence of specific markers of activation. However, increases in FAP expression in human dermal fibroblasts and human primary mammary fibroblasts after treatment with conditioned medium from breast cancer cells [34] and elevated FAP expression in NFs after treatment with conditioned medium from the colorectal cancer cell line HCT116 [47] have been reported. The stable activated state of NFs typical of co-cultures of HT29+NFs in our study is consistent with the study by Peng et al., where the co-cultivation of NFs with HT29 cells resulted in the highest FAP and aSMA expression at Day 4 of co-cultivation among six cancer cell lines [48]. The long duration of the activated state of the NFs and their elevated expression of activation markers in our experiments correlate with higher expression of integrin αvβ6 (similar expression level to the *ITGAV* gene and 12.2-fold, *p-adj* < 0.05 for the *ITGB6* gene) and of E-cadherin (12.5-fold, *p-adj* < 0.05) in HT29 cells compared to SW480. The mechanistic relationship of integrin αvβ6 expression in cancer cells with the fibroblast activation was revealed in the study by Peng et al. [48]. HT29 cells secreted inactive transforming growth factor β (TGF-β) and also expressed integrin αvβ6, which subsequently activated the TGF-β. Then, the activated TGF-β induced morphological changes in inactive fibroblasts and elevated the expression of activated fibroblast markers such as α-smooth muscle actin (α-SMA) and fibroblast-activating protein (FAP). The association between the activation of normal human dermal fibroblasts and E-cadherin and EpCAM expression in cancer cells was also shown by Eberlein et al. [49] for different non-small cell lung cancer (NSCLC) cell lines. The connection between the elevated expression of E-cadherin and the activation of fibroblasts could be explained by the contribution of E-cadherin to cell–cell contacts, which could result in more effective transmembrane signaling.

It should be noted that immunohistochemical staining of colorectal and breast cancer specimens has demonstrated the elevated expression of FAP and aSMA mainly in the interstitial/border zone of tumors, while in the tumor core, the expression of these markers is less pronounced [47,50]. Our in vitro model reflects the process of recruitment of NFs into the tumor. This occurs at the tumor border, and therefore, it is rational that FAP expression in NFs in co-cultures was higher at the beginning of the interaction with cancer cells than that in tumor-derived CAFs, and that it decreased with time. Taken together, the results enable us to suggest that the transformation of NFs into CAFs is a dynamic process, which agrees with the concept of fibroblast plasticity [5,37]. The heterogeneity of CAFs has been confirmed by numerous studies; in particular, *FAP* or *ACTA2* (aSMA) up-regulation was demonstrated not to occur for all CAF populations within a single tumor [51].

In our experiment with the use of RNA-seq analysis, the expression of podoplanin (*PDPN*), one of the CAF markers [38], was detected only in CAFs, but not in either NF monocultures or co-cultures of NFs with cancer cells. Recently, the PDPN expression in CAFs has been associated with poor prognosis for lung [52,53] and pancreatic cancers [54]. For colorectal cancer, it has been demonstrated that PDPN-positive CAF phenotype was associated with less aggressive tumors, whereas PDPN-low/α-SMAhigh or PDPN-low/S100A4high CAFs were associated with tumor progression [17]. The expression of several other CAF markers (*ASPN*, *CXCL10*, *THY1*) was detected only in CAFs but not in co-cultivated fibroblasts, which indicates the phenotypic difference between these cells. We assume that the in vitro activated normal fibroblasts in our study represent fibroblasts recently recruited to the tumor stroma rather than the resident CAFs of advanced tumors.

### 3.2. Colorectal Cancer Cells, Normal Fibroblasts, and Their Co-Cultures Remodel Collagen in Different Modes

The activation of fibroblasts normally occurs in response to tissue injury and inflammation. The main result of the activation is specific extracellular matrix remodeling: increased synthesis of ECM components such as collagen, fibronectin, elastin, and others, plus their deposition and remodeling [41]. Similar processes have been observed in tumors where they lead to a gradual stiffening of the tumor stroma due to increased collagen deposition and to an increased thickness, length, and alignment of the collagen fibers [13,14,18,55,56,57,58,59].

Stromal transformation is a complex process that encompasses a variety of chemical and physical mechanisms and processes. At least three main cell types participate in this phenomenon: cancer cells, CAFs, and tumor macrophages [60]. The function of CAFs is of the utmost importance and is under active consideration by many researchers [7,51,58]. Although the ability of cancer cells themselves to remodel the ECM has also been widely demonstrated, it has not yet been completely studied [13,15,16,61,62]. One of the few examples where the interactions of cancer cells with ECM were studied in detail is a series of publications that investigate the interactions of different types of breast cancer cells with fibronectin both in vitro and in vivo [63,64,65]. These studies show that more-mesenchymal breast cancer cells form the invasive front during tumor cell dissemination by fibronectin fibril alignment and that this alignment is transglutaminase-2 mediated. At the same time, the secondary metastatic tumors consist of more-epithelial cancer cells with low fibronectin expression levels, and the survival of more-epithelial cells is more-mesenchymal cell-dependent [64]. It was also shown that transglutaminase 2 and fibronectin are capable of accumulating in extracellular vesicles [65] and that these vesicles induce fibroblast-mediated fibronectin fibril alignment [63].

In turn, our study has clearly demonstrated the ability of colorectal cancer cells with different epithelial/mesenchymal states to induce collagen alignment in vitro. Using the optical method of SHG microscopy, we showed both qualitatively and quantitatively that cancer cells with a mesenchymal phenotype (SW480) are able to form thick, oriented collagen fibers, unlike the non-invasive HT29 cell line. These findings are consistent with early observations that in prostate and breast cancers, the more invasive cancer cell lines show higher average fibril fractions and greater collagen compaction levels, with densely packed fibers when compared to non-invasive cell lines [62,66]. In addition, it was recently shown that mesenchymal breast cancer cells enhance the accumulation of fibronectin, the glycoprotein of the extracellular matrix, in the presence of activated fibroblasts. Moreover, the correlation of this process with the promotion of metastasis was demonstrated [63,64]. Our transcriptome analysis revealed that more genes related to collagen biosynthesis and positive regulation and to collagen catabolism and negative regulation are active in the SW480 cell line compared to HT29. Moreover, in spite of the fact that more genes related to collagen fibril organization and ECM aggregation were active in cell line HT29 compared to SW480, two important collagen-remodeling genes, *LOXL2* and *LOXL3* (lysyl oxidases), were more active in SW480s. The *LOXL2* gene is considered as a promising target for cancer therapy [4]. The participation of *LOXL2* in collagen maturation [67] and the ability of SW480 cells to remodel ECM, as demonstrated in our above experiments, make it possible to consider it as an important participant in the formation of the border collagen structure typical of cancers.

We simulated in vitro the recruitment of NFs to the neoplastic process and demonstrated that NFs and cancer cell cross-talk resulted in the establishment of a dense ECM formed by large highly oriented fibers. The collagen structure in the co-cultures differed from those in both corresponding cancer cell monocultures and CAFs. Notably, this structure is quite similar to those described for the tumor boundaries in ex vivo and in vivo studies [13,18,60,68]. For CAFs, the predominance of proteolytic processes leading to a decrease in collagen content but with the preservation of highly oriented large collagen fibers was demonstrated in vitro. All these data support the idea of a graduated development of the tumor stroma, as proposed recently by Emon et al. [60]. The authors suggest that at the first stages of tumor formation, cancer cells recruit and activate fibroblasts, which, in turn, facilitate EMT in cancer cells through growth factors and ECM stiffening. In the further stages, ECM degradation processes are activated in tumors that promote evasion and metastasis. Our transcriptomic data also support this hypothesis, since we observed the activation of collagen remodeling processes and the suppression of collagen catabolic processes in cancer cells, and the activation of collagen biosynthesis, collagen fibril organization, and ECM aggregation processes in fibroblasts as a result of co-cultivation. In addition, transcriptome analysis demonstrated an increased activity of collagen biosynthesis and organization genes in normal fibroblasts compared to CAFs, which could explain the more active ECM degradation by CAFs compared to NFs, since the expression of the collagen catabolism-related genes in these cells is similar. Thus, according to the characteristics of ECM remodeling, normal fibroblasts co-cultured with cancer cells are more consistent with recently recruited stromal cells, while patient-derived CAFs represent the stromal cells of advanced tumors.

The in vivo data obtained in our experiment are pilot and require further investigation. However, we have highlighted that the most invasive cell line with a mesenchymal phenotype, SW480, failed to initiate tumors in three cases out of four upon conventional s.c. injection into the thigh. Although this cell line is tumorigenic in mice, the tumors are characterized by an extremely slow growth and require specific conditions for inoculation (such as the use of Matrigel and/or localization in the anterior regions of the trunk) [69]. HT29, which showed a pronounced epidermal phenotype and low cell motility in vitro, induced the densest tumors, where the collagen fibers were larger and more highly oriented compared to those in HCT116 tumors. This could be related to the ability of HT29 cells to induce and support an activated state of recruited NFs, as was demonstrated in vitro. In turn, this leads to the formation of the specific ECM structure typical of co-cultivated NFs, which was also demonstrated in our results reported above. Previous reports have indicated the unexpected discrepancy between the low in vitro invasiveness and high in vivo invasiveness and metastatic activity of the HT29 line [70].

These observations and numerous data about the ECM-mediated promotion of the invasive and metastatic potential of cancer cells underlie the need for further investigations in this field in order to discover new therapeutic targets and drug candidates to interfere with the cancer cell–fibroblasts interplay.

## 4. Materials and Methods

### 4.1. Cell Cultures

Five human colorectal adenocarcinoma cell lines (HT29, HCT116, SW480, Caco-2, and SW837, all lines obtained from the Ivanovsky Institute of Virology, Moscow), human skin fibroblasts, and tumor-derived cancer-associated fibroblasts (CAFs) were used in the study. The colorectal human adenocarcinoma cell lines were routinely grown in Dulbecco’s modified eagle’s medium (DMEM; Gibco, Life Technologies, Carlsbad, CA, USA) supplemented with 10% fetal bovine serum FBS (HyClone, Logan, UT, USA), 2 mM glutamine (PanEco, Moscow, Russia), 10 mg/mL penicillin, and 10 mg/mL streptomycin at 37 °C in a 5% CO_2_ humidified atmosphere. The normal fibroblasts and CAFs were maintained in the same conditions as the cancer cells. Cells were routinely harvested at 80% confluence using 0.025% trypsin-EDTA (Gibco, Life Technologies, Carlsbad, CA, USA).

### 4.2. Co-Culturing Cancer Cells and Fibroblasts in a Collagen-Based 3D Model

Normal human fibroblasts (NFs) and colorectal cancer cell lines were used to develop a three-dimensional tumor model based on type I rat tail collagen. The collagen type I was derived from rat tails according to the standard protocol [71]. To obtain a three-dimensional collagen matrix model, we mixed 3.5 volumes of collagen solution (1.5 mg/mL) with one volume of a reagent mixture (10x Medium 199 (Gibco, Life Technologies, Carlsbad, CA, USA), NaOH, Na2CO3, glutamine and 1x 4-(2-hydroxyethyl)-1-piperazineethanesulfonic acid (HEPES)). For obtaining the co-culture, the tumor cells and fibroblasts were mixed in a ratio of 1:1.2. Then, a suspension of the cells was mixed with the collagen gel in the ratio of 1:10. The total cell concentration was 1 × 10^5^ cells/mL with a final concentration of collagen gel of 1.2 mg/mL. Then, 500 μL of the mixture was immediately pipetted and added to a 35 mm^2^ glass-bottom dish and incubated for 30 min at 37 °C. Then, 2 mL of culture medium were added and routinely changed once every two days. Monocultures of each cell line, including the CAFs, were cultured under the same conditions and used as controls. Observations were made on days 1, 2, 5, and 7 of cultivation.

### 4.3. Isolation of CAFs and Normal Fibroblasts

The colon tumor samples were provided by the Volga Regional Medical Center (Nizhny Novgorod, Russia), while healthy skin samples were obtained from the Nizhny Novgorod Regional Children’s Clinical Hospital in accordance with protocols approved by the local ethical committee (approval No. 6, 17 April 2019). The samples of tumor-associated fibroblasts (CAFs) from patient tumor samples and normal human skin fibroblasts from healthy skin were obtained using standard techniques with minor modifications [72]. Clinically, the tumor samples (*n* = 3) represented moderately differentiated colorectal adenocarcinomas, which were classified as stage III or IV (TMN) (Appendix A). The tissue samples (tumor or skin) were repeatedly washed with culture medium with 10-fold antibiotic content until transparency of the supernatant was achieved. After that, the samples were crushed with sterile scissors. Then, enzymatic digestion was performed using Liberase™ (Roche, Belmont, CA, USA) as a dissociating agent in DMEM/F12 (Gibco, Life Technologies, Carlsbad, CA, USA) with the addition of hyaluronidase for 15 min at 37 °C and constant stirring. Then, fresh medium was added for enzymes neutralization, and the suspension was centrifuged at 1.6 × 10^3^ rpm for 7 min. The supernatant was removed, and the cell pellet with tissue pieces was resuspended in ACK Lysing Buffer (Gibco, Life Technologies, Carlsbad, CA, USA) according to the manufacture’s protocol to remove any red blood cells. Then, the cells were extensively washed, resuspended, and centrifuged in fresh DMEM medium. Then, the cell suspension was placed into 25-cm^2^ culture flasks. Tissue pieces were removed after fibroblast adhesion.

### 4.4. Immunofluorescence

For immunofluorescence analysis, the monocultures and co-culturing cells were seeded on 35 mm^2^ glass-bottom dishes covered with type-1 rat tail collagen in a growth medium. After 5 days of culturing, the expression of EMT markers and fibroblast-activated proteins was assessed by an immunocytochemical method using antibodies to E-cadherin (ab15148, Abcam, Cambridge, MA, USA), vimentin (ab16700, Abcam, Cambridge, MA, USA), FAP (ab28244, Abcam, Cambridge, MA, USA), αSMA (ab5694, Abcam, Cambridge, MA, USA), EpCam (ab20160, Abcam, Cambridge, MA, USA), and secondary antibodies conjugated with a fluorescent label, either Fluorescein isothiocyanate (FITC; ab6825, Abcam, Cambridge, MA, USA) or Alexa (ab6825, Abcam, Cambridge, MA, USA). Staining was performed according to the antibody manufacturer’s protocols, while the fluorescent dye DAPI was used for staining the nuclei. Fluorescence was recorded using a Leica DMIL fluorescence microscope (Leica, Wetzlar, Germany) equipped with filters: A4 UV BP 360/40 400 BP 470/40 for DAPI, CFP ET YFP ET (Ex: BP 500/20, Em: BP 535/30) for FITC and TX2 green BP 560/40 595 BP 645/75 for Alexa.

### 4.5. Wound-Healing Assay

Cancer cells were cultured for four days in six-well plates to confluence, and wounds were created with 10-μL pipette tips. Next, the cells were incubated as described above, and bright-field microscopic images of the wounds were taken from five randomly selected fields at 0, 48, and 72 h.

### 4.6. Spheroid Formation

To generate spheroids, cancer cells were seeded (100 cells per well) into a 96-well ultra-low attachment round-bottom plate (Corning, NY, USA). The plates were incubated (37 °C, 5% CO_2_) for 7 days. Culture medium was gently changed every three days. Spheroid formation was monitored with an inverted light microscope (Leica, Wetzlar, Germany).

### 4.7. Tumor Xenografts

The protocol for our animal experiments was approved by the Ethical Committee of the Privolzhsky Research Medical University (Russia). Experiments were performed on female, athymic, nude mice of 20–22 g body weight, purchased from the Pushchino animal nursery (Pushchino, Russia). To generate tumors, HT29, HCT116, or SW480 cells were injected subcutaneously in the left flank at a dose of 5 × 10^6^ cells in 200 μL PBS.

### 4.8. Fluorescence-Activated Cell Sorting

In order to separate co-cultured cancer cells and NFs, the NFs were stained, before co-culturing, with 2 µM 5-Chloromethylfluorescein diacetate (CMFDA; Sigma-Aldrich, St. Louis, MO, USA) in serum-free medium for 20 min; then, they were washed in complete medium and mixed with the cancer cells. After 5 days of co-culturing, the cells were detached with trypsin solution and counterstained with 10 µM propidium iodide for 10–30 min to discriminate dead cells. Live NFs and cancer cells were sorted with a FACSAria III sorter (BD Biosciences, Franklin Lakes, NJ, USA) into 15 mL falcon tubes filled with 5 mL of complete medium. Aliquots (5%–10%) of sorted cells were re-analyzed to check for sorting purity and were found to be in the range 97.5%–99.8%. After sorting, the cells were washed once with PBS and processed for RNA isolation.

### 4.9. RNA Isolation and Sequencing

At least 250,000 cells were collected by centrifugation and lysed using ExtractRNA (Evrogen, Moscow, Russian Federation). The cell lysates were stored at −70 °C until RNA extraction. RNA was extracted with RNA Clean & ConcentratorTM-5 (Zymo Research, Irvine, CA, USA). DNase I treatment was performed in-column manner during the clean-up. The quality of the extracted RNA was determined using TapeStation 2200 (Agilent Technologies, Santa Clara, CA, USA) with Agilent High Sensitivity RNA ScreenTape (Agilent Technologies). Since the RNA Integrity Number (RINe) was greater than 7 for all samples, poly(A) RNA isolation was performed with a NEBNext Poly(A) mRNA Magnetic Isolation Module (New England BioLabs, Ipswich, MA, USA). The NEBNext^®^ Ultra II Directional RNA Library Prep Kit for Illumina^®^ with Sample Purification Beads (New England BioLabs) and a NEBNext^®^ Multiplex Oligos for Illumina (Index Primers Set 1–3) (New England BioLabs) were used for the preparation of single-indexed libraries. The preparation of the libraries was performed in technical triplicate for HT29, SW480, NF, HT29^NF^, SW480^NF^, NF^HT29^, and NF^SW480^. For CAF1 and CAF2, the preparation of the libraries was performed in technical duplicate. After the preparation, the individually indexed libraries were mixed in equimolar amounts; the final mixture was analyzed with TapeStation using Agilent High Sensitivity DNA ScreenTape (Agilent Technologies). The sequencing was performed with an Illumina NovaSeq 6000 System (Illumina, San Diego, CA, USA, SP-type, single-end reads, 100 bp).

The quality of raw reads was assessed using the MultiQC tool [73]. Since the quality of the reads in all samples was acceptable, trimming was not performed. The raw reads were mapped on the human reference genome (hg38) using HISAT2 (Galaxy Version 2.1.0+galaxy5; [74]). The featureCounts (Galaxy Version 1.6.4+galaxy1; [75]) was used for counting reads to GENCODE release 33 annotated genes [76]. Mitochondrial genes were removed prior to analysis. The obtained read counts were analyzed with DESeq2 (Galaxy Version 2.11.40.6+galaxy1 [77]) and also were converted into transcripts per kilobase million (TPM) values. Then, the TPM values were averaged over the replicates. The data discussed in this publication have been deposited in NCBI’s Gene Expression Omnibus [78] and are accessible through GEO Series accession number GSE15534 (https://www.ncbi.nlm.nih.gov/geo/query/acc.cgi?acc=GSE155343).

### 4.10. Second Harmonic Generation (SHG) Microscopy

The visualization of fibrillar collagen and the quantitative assessment of its biophysical properties in vitro and in vivo was performed using second harmonic generation (SHG) microscopy, the established optical method based on the ability of centrosymmetric structures, such as collagen fibers, to reconvert two-photon excitation to form newly emitted photons with half of the wavelength [79].

For SHG and two-photon fluorescence microscopy in vitro, a multiphoton tomograph MPTflex (JenLab GmbH, Jena, Germany) equipped with a tunable 80 MHz, 200 fs Ti:Sa laser MaiTai (Spectra Physics, Santa Clara, CA, USA) was used. The objective lens was a Plan-Apochromat 40×/1.3 oil (Carl Zeiss, Oberkochen, Germany). The SHG signal from the collagen fibers was excited at a wavelength of 750 nm and detected in the range 373–387 nm in the backward direction. Two-photon fluorescence was excited at a wavelength of 750 nm and detected in the range 410–660 nm. The average power applied to the sample was ≈12 mW.

The in vivo SHG and two-photon fluorescence images were acquired using a LSM 880 laser scanning microscope (Carl Zeiss) equipped with a tunable 80 MHz, 140 fs Ti:Sa laser MaiTai HP (Spectra Physics). The images were acquired with a C Plan-Apochromat 40x/1.3 NA oil immersion objective through a mounted coverslip that was placed on the tumor surface. The SHG signal from the collagen fibers was excited at 800 nm and detected in the range 371–421 nm in the backward direction. Two-photon fluorescence was excited at a wavelength of 800 nm and detected in the range 433–660 nm. The average power applied to the sample was ≈9 mW.

SHG imaging was implemented in vivo on the 14–15th day of tumor growth. Before the imaging procedures, the mice were anesthetized intramuscularly with a mixture of Zoletil (40 mg/kg, 50 μL, Virbac SA, Carros, France) and 2% Rometar (10 mg/kg, 10 μL, Spofa, Prague, Czech Republic), and a skin flap over the tumor was surgically opened. Once imaging was completed, the animals were sacrificed by cervical dislocation, and the tumors were excised for histopathological verification.

### 4.11. Quantitative Analysis of Collagen

To evaluate the amount of collagen and its local organization, the following parameters of the SHG signal were calculated: the mean intensity, the density, the median, the non-uniformity, kurtosis, skewness, coherency, and energy [80].

The mean value is the simplest first-order statistical parameter used to evaluate the collagen SHG signal, and it correlates with the amount of collagen. The higher the mean value, the more collagen in the ROI. Density is also a characteristic of the amount of collagen in the SHG image. This parameter is normalized to the ROI area; therefore, it does not depend on the image size. The median parameter shows the most common value of the number of collagen fibers in the field of view. The non-uniformity parameter (S) is calculated as the ratio of the standard deviation to the mean value of the SHG signal. A higher S value indicates a more heterogeneous distribution of the collagen. Skew (histogram asymmetry) is a third-order moment that determines the degree of symmetry of the gray distribution relative to the average. SHG images with thinner collagen fibers have higher asymmetry. Kurtosis is the fourth-order moment that defines the difference between the observed distribution in the histogram and a Gaussian distribution. The greater the value of the kurtosis parameter, the brighter the fibers. Energy (uniformity) shows the degree of variability of the SHG signal in the image. The higher the value, the more uniform the collagen structure. Coherence is a measure of orderliness or the presence of a dominant fiber direction. The larger the value, the more ordered the collagen fibers (there is a specific direction). The value of the coherency is in the range 0–1, where 1 corresponds to a highly oriented structure, and 0 corresponds to isotropic areas. A detailed description of these quantitative metrics and their calculations can be found in [80].

Quantitative analysis of the SHG signal was performed using ImageJ software (National Institutes of Health, Bethesda, MD, USA) and the OrientationJ plugin. At each time point, 3–31 ROIs (each about 2500 μm^2^) were assessed for each culture dish or tumor.

### 4.12. Statistical Analysis

The values are expressed as mean ± standard deviation (SD) or the standard error of the mean (SEM). To estimate the statistical significance of the differences, the ANOVA with Bonferroni post-hoc test or a two-tailed Student’s *t*-test were used where appropriate. The *p*-values ≤ 0.05 were considered significant.

## 5. Conclusions

The role of the stroma in promoting cancer growth and metastasis is now a commonly accepted concept [1,2,27]. Normal fibroblasts recruited by cancer cells are necessary participants in the tumor invasion front; however, the molecular mechanisms of the cancer–stroma crosstalk are far from being well studied. In this report, we tried to correlate the dynamic processes of collagen biosynthesis and remodeling and the transcriptome changes during cancer and stroma cell interactions. To this end, we explored the impact of the change in the phenotype of colorectal cancer cells from epithelial to mesenchymal on their ability to activate fibroblasts and to remodel collagen structure in parallel with changes in the corresponding collagen-related gene expression.

Two data lines were identified: Cancer cells impact on collagen structure and its change due to co-cultivation with fibroblasts Cancer cells, when cultured alone, differ in their collagen remodeling capacity. An increase in the density of collagen was observed only for the SW480 (invasive) cell line, but not for HT29. SW480 demonstrated the ability to create quite thick, oriented fibers.Co-culturing of colorectal cancer cell lines with normal fibroblasts resulted in a greater amount of fibrillar collagen during all periods of cultivation.Fibroblast-mediated remodeling of collagen led to the formation of highly ordered fiber networks, which are more favorable for cancer cell invasion. However, after prolonged cultivation in the absence of cancer cells, the CAFs lost the ability to organize collagen in ways that promote invasion, and they actually initiated matrix degradation.Expression of “collagen” genes in cancer cells and its change during co-cultivation with fibroblasts The genes related to collagen biosynthesis and catabolism tend to be more active in the more-mesenchymal and mobile SW480 cells than in the more-epithelial and less mobile HT29 cells. Moreover, the LOXL2 and LOXL3 genes, which are necessary for collagen fibril organization, are SW480 specific.The co-cultivation of colon cancer cell lines HT29 or SW480 with NFs leads to the activation of collagen biosynthesis and collagen fibril organization genes in both the fibroblasts and the cancer cells.The genes related to collagen biosynthesis and collagen fibril organization tend to be more active in skin fibroblasts (NFs) than in colon CAFs, whereas CAF marker genes are significantly up-regulated in colorectal CAFs compared to normal skin fibroblasts.The co-cultivation of the colorectal cancer cell lines HT29 or SW480 with skin fibroblasts leads to the activation of several CAF marker genes expression in the fibroblasts (ACTA2, POSTN, TNC, and DES). During co-cultivation, the HT29 line activates normal fibroblasts to a greater extent than does SW480.

Our findings suggest that normal fibroblasts, activated by cancer cells, strongly contribute to the organization of the extracellular matrix, thus being favorable for migratory potency. We also hope that further analysis of our transcriptomic data will help us find new therapeutical targets among the genes that mediate cancer cell-fibroblast interactions. In conclusion, targeting the ability of cancer cells to activate normal fibroblasts can be considered an important field in which to search for new therapeutic strategies.

## Figures and Tables

**Figure 1 ijms-21-08119-f001:**
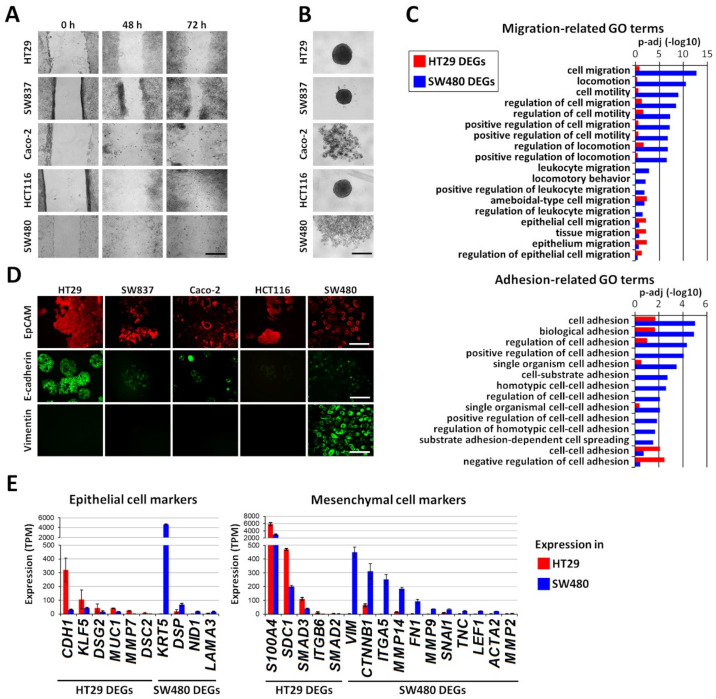
Identification of epithelial/mesenchymal state of five human colorectal cancer cell lines. (**A**) Monitoring of cell migration using wound-healing assay. Representative microscopic images of wound closure at 0, 48, or 72 h post wounding. Scale bar, 400 μm. (**B**) Spheroid formation ability of colorectal cancer cell lines. Cell lines were cultured on ultra-low attachment round-bottom plates, and the optical microscopy images were obtained 7 days after cell seeding. Scale bar, 400 μm. (**C**) Cell migration- and adhesion-related Gene Ontology Biological Process (GO BP) terms that are enriched in sets of differentially expressed genes between the HT29 and SW480 cancer cell lines. RNA-seq data were used to identify differentially expressed genes (|log2(FC)| >1) that were subjected to functional enrichment analyses using the DAVID analysis tool to identify enriched GO BP terms. (**D**) Immunofluorescence staining of the indicated cell lines for the epithelial cell adhesion molecule (EpCAM) (red) and the epithelial marker E-cadherin (green) and mesenchymal markers vimentin (green). Representative fluorescence microscopic images showing the epithelial phenotype of HT29 cells and the mesenchymal phenotype of SW480 cells. Scale bar, 400 μm. (**E**) Expression of epithelial and mesenchymal marker genes differentially expressed between HT29 and SW480 cancer cells. RNA-seq data were used to calculate the gene expression levels in transcripts per kilobase million (TPM). Mean values ± standard deviation in triplicate are shown.

**Figure 2 ijms-21-08119-f002:**
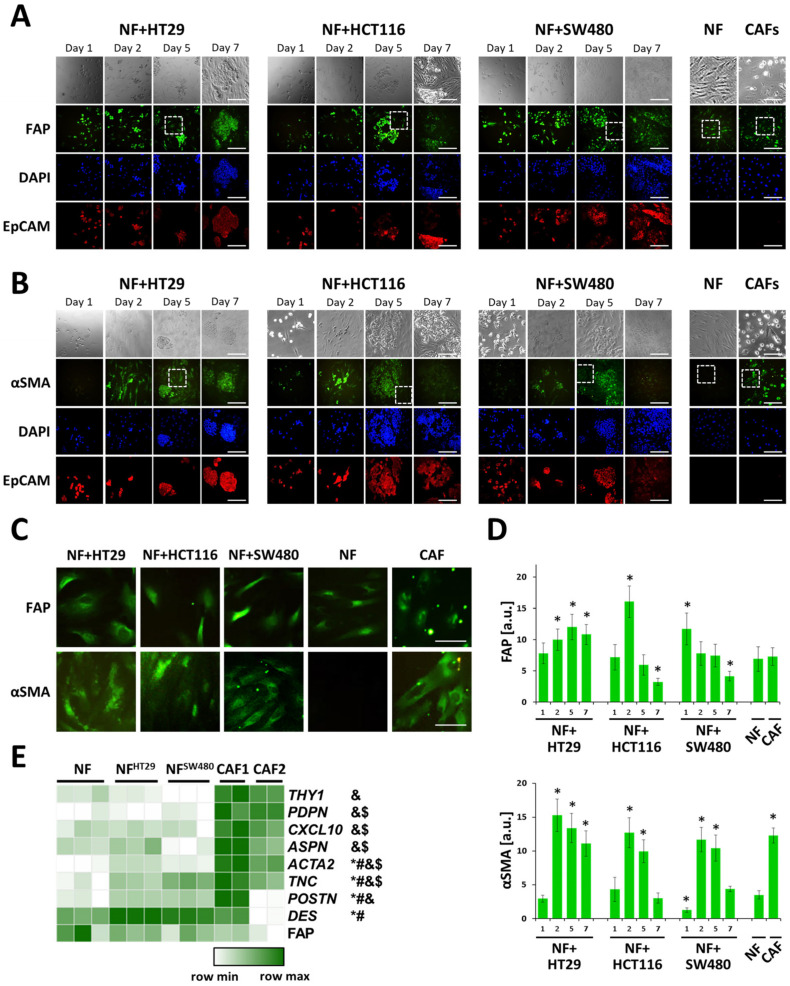
Activation of normal fibroblasts upon co-culturing with colorectal cancer cells. Immunofluorescence staining of the co-cultures of HT29, HCT116, or SW480 cells with normal fibroblasts (NFs) on days 1, 2, 5 and 7, NFs and patient-derived CAFs for (**A**) FAP, fibroblast activation protein α1, and (**B**) aSMA, α-smooth muscle actin. Cells were additionally stained against 4′,6-diamidino-2-phenylindole (DAPI; nucleus; blue) and EpCAM (epithelial cell adhesion molecule; red). Scale bar, 400 μm (**C**) A higher magnification of the areas corresponding to the dashed square in (**A**), (**B**). Scale bar, 100 μm. (**D**) Quantification of FAP and aSMA staining (averaged fluorescence intensity of fibroblasts, at least eight regions of interest (ROIs) per condition with 2–5 cells in each ROI). Mean ± SD. *, *p* < 0.05 from NF. (**E**) Gene expression heatmap of cancer-associated fibroblast (CAF) markers with differential activity in various types of fibroblasts. NFs—normal skin fibroblasts, NF^HT29^—normal fibroblasts isolated from co-cultures with HT29 cancer cells on day 5, NF^SW480^—normal fibroblasts isolated from co-cultures with SW480 cancer cells on day 5, CAF1-2—colon cancer-associated fibroblasts (cultures 1 and 2). *p-adj* < 0.05 for gene expression in NFs compared to *, NF^HT29^; #, NF^SW480^; &, CAF1; $, CAF2.

**Figure 3 ijms-21-08119-f003:**
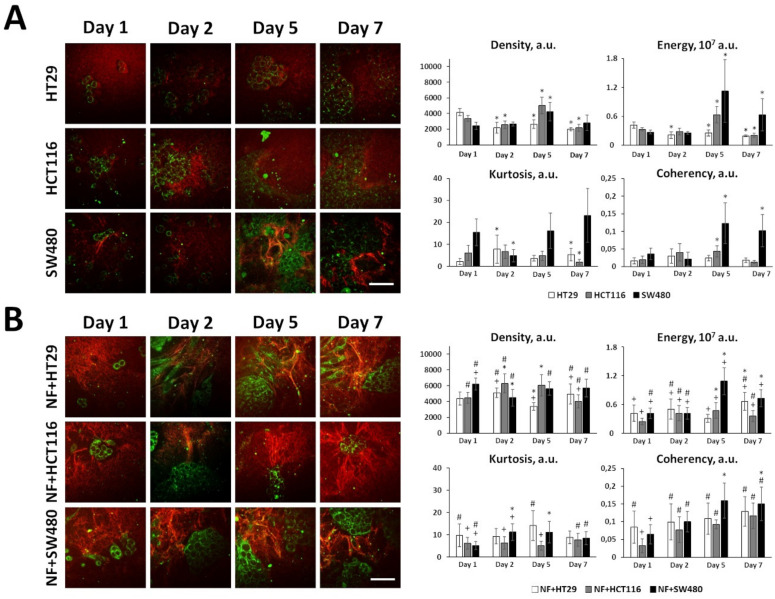
In vitro fibrillar collagen assessment in collagen-based 3D models using second harmonic generation (SHG) microscopy. The collagen structure in 3D systems containing cancer cells and/or fibroblasts in collagen gel was monitored with SHG microscopy and then quantified using the parameters of the SHG signal. Right: Representative SHG images (red) combined with two-photon excited fluorescence (TPEF, green) images of monocultures of cancer cells (**A**), co-cultures of cancer cells and normal fibroblasts (**B**), and monocultures of normal fibroblasts (NFs) and patient-derived CAFs (**C**). Bar is 50 µm, applicable to all images. Left: Several quantitative parameters of the SHG signals including density, energy, kurtosis, and coherency. Mean ± SD. Imaging was performed at 1, 2, 5, and 7 days of culturing cells in collagen gel. Key: *, *p* < 0.05 from Day 1; #, *p* < 0.05 from corresponding monoculture of cancer cells on the same day, +, *p* < 0.05 from NFs on the same day, &, *p* < 0.05 from CAFs on the same day. All values are given in arbitrary units (a.u.). All quantitative parameters of the SHG signal and *p*-values are given in Appendix A.

**Figure 4 ijms-21-08119-f004:**
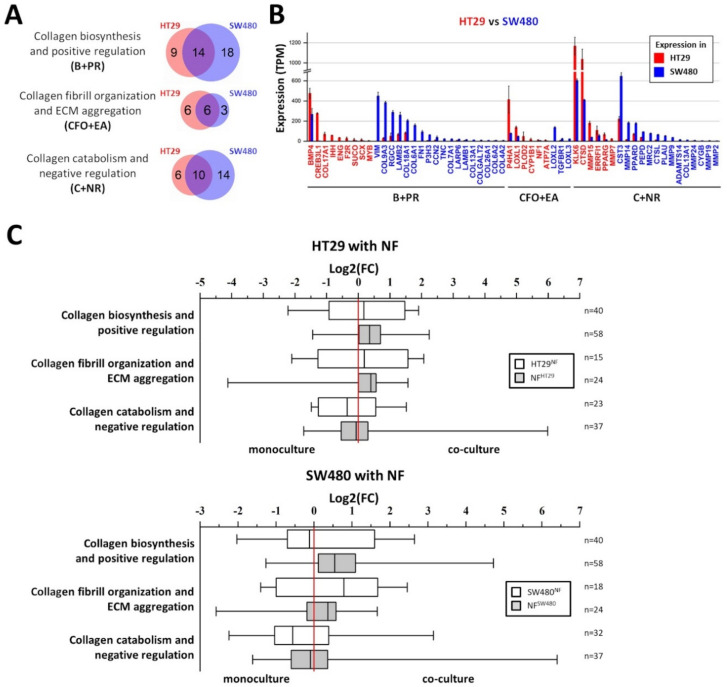
Expression of collagen-related genes in cancer cells and its changes as a result of the co-cultivation of cancer cells with normal fibroblasts. (**A**) Venn diagrams showing the overlap between significantly expressed collagen-related genes (*p-adj* < 0.05; TPM ≥ 5) in colorectal cancer cell lines HT29 and SW480 in monoculture. Three gene categories are shown: collagen biosynthesis and positive regulation (B+PR), collagen fibril organization and extracellular matrix (ECM) aggregation (CFO+EA), and collagen catabolism and negative regulation (C+NR). (**B**) Histogram showing the expression of collagen-related genes, differentially expressed (*p-adj* < 0.05; TPM ≥5; │log2(FC)│ ≥ 1) in HT29 (red gene name) and SW480 (blue gene name). Red and blue bars represent the expression of genes in HT29 and SW480, respectively. Mean values (*n* = 3) are shown in TPM, error bars represent the standard deviation (*n* = 3). (**C**) Box plots representing changes of collagen-related gene expression as a result of co-cultivation of HT29 or SW480 cell lines with normal fibroblasts (NFs). The distributions of the log2(FC) values for significantly expressed collagen-related genes (*p-adj* < 0.05; TPM ≥ 5) in 3 gene categories (B+PR, CFO+EA, and C+NR) are shown; that is, the expression of the gene in co-culture divided by the expression of the gene in monoculture. Thus, gene expression that is up-regulated during co-cultivation has log2(FC) > 0, and gene expression that is down-regulated during co-cultivation has log2(FC) < 0. The twenty-fifth, fiftieth, and seventy-fifth percentiles are used; whiskers show the minimal and maximal log2(FC)-values. White boxes represent the changes of collagen-related gene expression in the cancer cells (HT29 or SW480); gray boxes represent the changes of collagen-related gene expression in the NFs. The red line indicates log2(FC) = 0, that is, no change in gene expression. The number of genes (*n*) used for each box is indicated on the right side of the chart.

**Figure 5 ijms-21-08119-f005:**
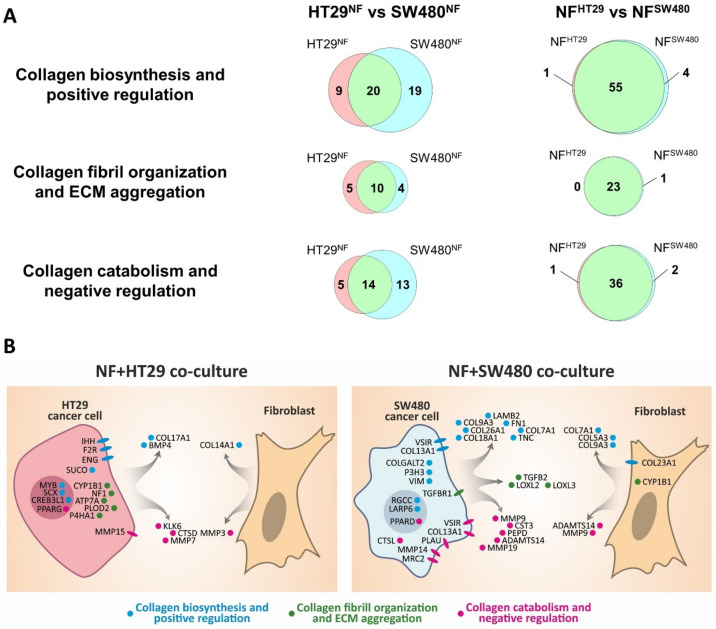
Differential expression of the collagen-related genes in co-cultivated cancer cells and fibroblasts. (**A**) Venn diagrams showing the overlap between significantly expressed collagen-related genes (*p-adj* <0.05; TPM ≥5) in co-cultivated colorectal cancer cell lines (HT29^NF^ and SW480^NF^) and normal fibroblasts (NF^HT29^ and NF^SW480^). Three gene categories are shown: collagen biosynthesis and positive regulation, collagen fibril organization and ECM aggregation, and collagen catabolism and negative regulation. (**B**) Scheme showing localization of collagen-related proteins that can determine differences in collagen remodeling by different co-cultures. Collagen-related differentially expressed genes (*p-adj* <0.05; TPM ≥5; │log2(FC)│≥1)) were determined between cancer cells and between fibroblasts of both co-cultures. Some SW480 DEGs encoding extracellular proteins were excluded because of their much higher expression in fibroblasts (log2(FC) ≥5).

**Figure 6 ijms-21-08119-f006:**
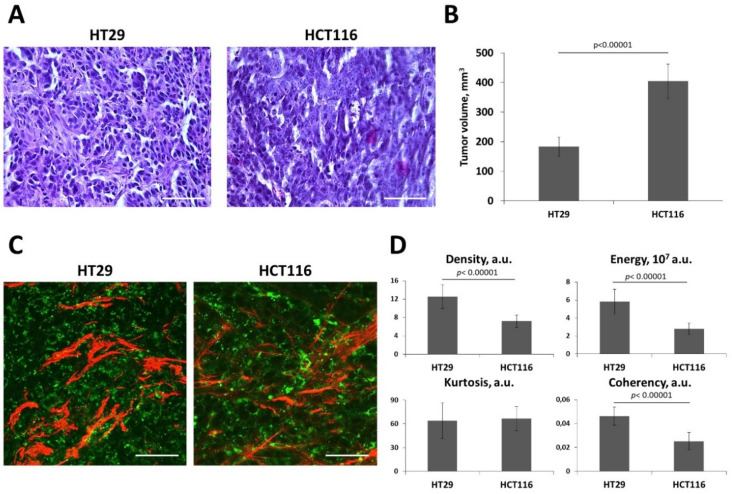
In vivo collagen structure in HT29 and HCT116 tumor xenograft. (**A**) Histopathology images on the 14th day of tumor growth. Hematoxylin and eosin staining. Scale bar, 50 μm. (**B**) Tumor volume (V) measurements on 14th day of growth. Mean ± SEM. (**C**) Representative SHG images (red, collagen fibers) combined with two-photon excited fluorescence images (TPEF, green, cellular autofluorescence) of tumors. Image size is 212 × 212 μm. Scale bar, 50 μm. (**D**) Quantitative parameters of the SHG signal including density, energy, kurtosis, and coherency. Mean ± SEM, (*n* = 3–4 tumors, at least 15 fields of view in each tumor). Cancer cells were implanted subcutaneously in nude mice. Imaging was performed on Day 14 of tumor growth. SHG images and quantitative analysis of the SHG signal show a denser and more oriented collagen structure in the HT29 compared with HCT116 tumors. *p*-values denote significant differences.

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
