# Peer review of "Expression of EMT-Related Genes in Hybrid E/M Colorectal Cancer Cells Determines Fibroblast Activation and Collagen Remodeling"

_ijms, 2020, doi:10.3390/ijms21218119_

Round 1
Reviewer 1 Report
In this study the authors investigated the processes of collagen biosynthesis and remodeling in parallel with the transcriptome changes during cancer cells and fibroblasts interactions. The authors demonstrate that targeting the ability of epithelial cancer cells to activate normal fibroblasts can provide a new anticancer therapeutic strategy. The studies are nicely executed, and most of the findings are straight forward. While the concept of the work is very interesting and informative, there are a few concerns that the authors need to address in their study.
- Please provide significance of the research in more details.
- Please provide rationale for using mentioned cell lines in the study.
- Please improve discussion section
- Please cite the following articles in your manuscript
- Transglutaminase-2 facilitates extracellular vesicle-mediated establishment of the metastatic niche
- Spleen Tyrosine Kinase–Mediated Autophagy Is Required for Epithelial–Mesenchymal Plasticity and Metastasis in Breast Cancer.
- Autocrine fibronectin inhibits breast cancer metastasis.
- Fibronectin expressing mesenchymal tumor cells promote breast cancer metastasis
- Pyruvate carboxylase supports the pulmonary tropism of metastatic breast cancer
- Regulation of epithelial-mesenchymal transition and metastasis by TGF-β, P-bodies, and autophagy
- The Dynamic Relationship of Breast Cancer Cells and Fibroblasts in Fibronectin Accumulation at Primary and Metastatic Tumor Sites
- Dynamic transition of the blood-brain barrier in the development of non-small cell lung cancer brain metastases
- Inhibition of pyruvate carboxylase by 1α, 25-dihydroxyvitamin D promotes oxidative stress in early breast cancer progression
- Understanding ECM-Based Drug Resistivity in Breast Cancer
Author Response
Dear Reviewer,
Below are the responses to Your questions point by point. The questions are underlined. We found Your comments helpful and tried our best to make all the amendments You recommended.
- Please provide significance of the research in more details.
The significance of the research is now provided in the Introduction section (see lines 54-59)
- Please provide rationale for using mentioned cell lines in the study.
We clarified the choice of cancer type in the Introduction section (see lines 103-107). The experiments were performed on the most frequently used and well characterized cell lines of this type of cancer according to the literature. Another important criterion for our choice was based on the difference in their epithelial/mesenchymal morphology to cover wide spectrum of phenotypes (see lines 121-123, first paragraph of the Results subsection 2.1 “Epithelial/mesenchymal states of cancer cells”).
- Please improve discussion section
The Discussion section was improved with help of the publications the reviewer indicated, some new actual information was also added – see lines 578-587 and 595-598, subsection 3.2 “Colorectal cancer cells, normal fibroblasts and their co-cultures remodel collagen in different modes”
- Please cite the following articles in your manuscript:
- Transglutaminase-2 facilitates extracellular vesicle-mediated establishment of the metastatic niche
- Spleen Tyrosine Kinase–Mediated Autophagy Is Required for Epithelial–Mesenchymal Plasticity and Metastasis in Breast Cancer.
- Autocrine fibronectin inhibits breast cancer metastasis.
- Fibronectin expressing mesenchymal tumor cells promote breast cancer metastasis
- Pyruvate carboxylase supports the pulmonary tropism of metastatic breast cancer
- Regulation of epithelial-mesenchymal transition and metastasis by TGF-β, P-bodies, and autophagy
- The Dynamic Relationship of Breast Cancer Cells and Fibroblasts in Fibronectin Accumulation at Primary and Metastatic Tumor Sites
- Dynamic transition of the blood-brain barrier in the development of non-small cell lung cancer brain metastases
- Inhibition of pyruvate carboxylase by 1α, 25-dihydroxyvitamin D promotes oxidative stress in early breast cancer progression
- Understanding ECM-Based Drug Resistivity in Breast Cancer
Thank You for Your suggestions. We found them very valuable and relevant, and their citation helped us to improve the Discussion section (see lines 578-587, second paragraph of the Discussion subsubsection 3.2 “Colorectal cancer cells, normal fibroblasts and their co-cultures remodel collagen in different modes”).
Reviewer 2 Report
This manuscript contains valuable data and shows novel ideas and approaches with new techniques.
However, it needs to improve some points.
1) First, it is not clear how to develop new therapeutic strategy from their findings. Please describe in detail.
2) Authors need to identify or explain what component might induce remodeling of collagen from cancer cells or fibroblast. What component is involved in communication between cancer cells and fibroblasts?
3) Please add some scheme for your proposed mechanism.
4) In case of tumour xenograft, authors need to show the data about cancer progression and some H&E staining, etc.
Author Response
Dear Reviewer,
Below are the responses to Your questions point by point. The questions are underlined. We found Your comments helpful and tried our best to make all the amendments You recommended.
We hope that the revised version is now suitable for publication.
- First, it is not clear how to develop new therapeutic strategy from their findings. Please describe in detail.
Several therapeutical strategies targeting tumor stroma were described (e.g., LOXL2 – ECM remodeling enzyme), however, they appeared to be not efficient enough (see Introduction, lines 54-59). We assume that detailed analysis of our transcriptomic data will be helpful in finding new therapeutical targets among the genes involved in cancer cell-fibroblast interactions (see Conclusions, lines 844-846). An example could provide immune checkpoint therapy. Here interaction of cancer ligands with immune cell receptors causes the inhibition of immune response. Antibodies destroying the receptor-ligand interaction allow to reanimate the anticancer immune response. Receptors and ligands come in many forms, but they all have one thing in common: a receptor recognizes just one (or a few) specific ligands, and a ligand binds to just one (or a few) target receptors. This is the basis of well recognized druggable properties of receptors and their cognate ligands, which makes them especially useful clinical targets. We hope that the analysis of the revealed interactions will allow us to find similar interacting pairs, the attacking of which could have a therapeutic anticancer effect. However, this analysis was out of scope of this study.
- Authors need to identify or explain what component might induce remodeling of collagen from cancer cells or fibroblast. What component is involved in communication between cancer cells and fibroblasts?
To answer the first part of this question, the following analysis was carried out. We consider that secreted proteins or proteins localized on the plasma membrane are directly involved in collagen remodeling. Thus differentially expressed proteins are main cause of demonstrated differences in collagen remodeling by two co-cultures. We allocated collagen-related genes that differ in expression between the cancer cells or fibroblast from two co-cultures (HT29NF vs SW480NF and NFHT29 vs NFSW480) according to the preferred localization of their products in cellular compartments, including the extracellular space (see added Figure 5B). We used Genecards database to identify corresponding protein localization. As the result of this analysis, we propose a scheme explaining which components produced by fibroblasts and cancer cells can determine the differences in the structure of the collagen matrix between the two co-cultures used in this study.
The second part of this question addresses the possible mechanisms of cancer cell-fibroblast interactions. This specific problem was out of scope of this study. However, we plan to investigate this problem in our future research. At present we have already identified a variety of possible interactions of cancer cells with fibroblasts (see the attached Figure 1) based on the databases provided in (Ramilowski et al., 2015, https://doi.org/10.1038/ncomms8866 and Cabello-Aguilar et al., 2020, https://academic.oup.com/nar/article/48/10/e55/5810485). The determination of key interactions among them is our next purpose.
Figure 1. Potential ligand-receptor interactions in co-cultures. The numbers of expressed (TPM ≥5) ligand (−>) and receptor (−<) genes, as well as numbers of possible interactions are shown for each co-culture.
- Please add some scheme for your proposed mechanism.
We added this scheme in Figure 5 B (see the description of our rationale in answer to 2nd question).
- In case of tumour xenograft, authors need to show the data about cancer progression and some H&E staining, etc.
The data about tumor size and histopathology have been added in the Manuscript (Figures 6A, 6B and lines 456-462, 476-479). Unfortunately, the animal study was performed during the Covid-19 pandemic, the access into the institute was restricted and we were forced to limit ourselves to only 14 days of growth.
Reviewer 3 Report
In this article, 'Expression of EMT related genes in hybrid E/M colorectal cancer cells determines fibroblast activation and collagen remodeling' author describes the ability of cancer cells to activate normal fibroblast for innate collagen remodeling capacity, thus providing a new therapeutic target. This article has some novel findings and can be accepted in the current form.
Author Response
Thank You for Your comments. We did our best to make all the amendments that other reviewers recommended. We hope that the revised version is also suitable for publication.
Round 2
Reviewer 1 Report
The authors have addressed all the comments.
Reviewer 2 Report
All issues from my part are cleared.